# Auditory experience controls the maturation of song discrimination and sexual response in *Drosophila*

**Xiaodong Li, Hiroshi Ishimoto, Azusa Kamikouchi\***

Graduate School of Science, Nagoya University, Nagoya, Japan

**Abstract** In birds and higher mammals, auditory experience during development is critical to discriminate sound patterns in adulthood. However, the neural and molecular nature of this acquired ability remains elusive. In fruit flies, acoustic perception has been thought to be innate. Here we report, surprisingly, that auditory experience of a species-specific courtship song in developing *Drosophila* shapes adult song perception and resultant sexual behavior. Preferences in the song-response behaviors of both males and females were tuned by social acoustic exposure during development. We examined the molecular and cellular determinants of this social acoustic learning and found that GABA signaling acting on the GABA$_A$ receptor Rdl in the pC1 neurons, the integration node for courtship stimuli, regulated auditory tuning and sexual behavior. These findings demonstrate that maturation of auditory perception in flies is unexpectedly plastic and is acquired socially, providing a model to investigate how song learning regulates mating preference in insects.

DOI: https://doi.org/10.7554/eLife.34348.001

## Introduction

Vocal learning in infants or juvenile birds relies heavily on the early experience of the adult conspecific sounds. In humans, early language input is necessary to form the ability of phonetic distinction and pattern detection in the phase of auditory learning (*Doupe and Kuhl, 1999*; *Kuhl, 2004*). Because of the strong parallels between speech acquisition of humans and song learning of songbirds, and the difficulties to investigate the neural mechanisms of human early auditory memory at cellular resolution, songbirds have been used as a predominant model in studying memory formation during vocal learning. In juvenile songbirds, a small subset of neurons in the higher-order auditory cortex responded selectively to a song experienced in the early exposure, and thus were thought to be the neuronal substrate for song memory formation (*Yanagihara and Yazaki-Sugiyama, 2016*). However, it remains unclear how the neurons that represent the sound memory are incorporated into the higher-order integration center to direct the sensorimotor output.

The auditory system of *Drosophila melanogaster* has attracted increasing attention in recent years for the huge progress in understanding its underlying neural mechanisms (*Clemens et al., 2015*; *Kamikouchi et al., 2009*; *Zhou et al., 2015*). The courtship song produced by wing vibration of males during the courtship ritual has been studied most among the communication sounds in flies (*Laturney and Billeter, 2014*). The courtship song, thought to be the primary cue affecting the female's choice of the courting male (*Crossley et al., 1995*; *Villella and Hall, 2008*), comprises two components: trains of pulses called pulse song and sequences of humming called sine song (*von Schilcher, 1976a*). Although the function of the sine song is not well understood, sound playback experiments have demonstrated that the pulse song promotes copulation in paired flies (*Kyriacou and Hall, 1982*; *Ritchie et al., 1999*). Receptivity of females is improved by playback of an artificial pulse song, which reduces female rejection responses and shortens the time to

**\*For correspondence:**
kamikouchi@bio.nagoya-u.ac.jp

**Competing interests:** The authors declare that no competing interests exist.

**eLife digest** Many mammals and birds have a critical period in youth when hearing the vocal cues of their parents helps them to learn the specific features of their communication sounds. Scientists have been studying the brains of humans, birds and other animals to find out what is happening in their brains when the animals hear these sounds. However, the brains of these species are too complex to fully understand how early vocal influences shape the brain networks that control behavior.

Therefore, scientists often use 'simpler' organisms, such as insects, to study these processes. For example, fruit flies use a series of courtship behaviors – including mating calls – to attract their potential mates. To produce a courtship song, males vibrate their wings, which consists of short pulsed songs and sequences of humming. The time interval between the pulses is specific to a species. Until now it was thought that these mating calls are innate behaviors that cannot be learned or modified.

To test this, Li et al. clipped the wings of male fruit flies so they could not produce their own song. First, they placed the females with the males and played one species-specific courtship song, and one from a different species. Both songs resulted in successful copulation and did not affect the female's choice.

To find out if a previous experience of a courtship song can influence the behavior of the fruit flies, Li et al. raised one group hearing their species-specific song and the other with a song from a different species. The results showed that females growing-up with their species-specific song, rejected males when a song of another species was played. However, the females accustomed to the other species' song did not change their song preference and receptivity towards males. The same was also true for males: male fruit flies raised with their species-specific song later ignored another species' song, which usually increased their mating drive.

Li et al. further identified a specific region in the brain of the fruit flies known to be important for courtship, and a key molecule that regulated this behavior. These findings suggest that far from being innate, the mating preference in fruit flies can be learned and influenced by social experience.

A next step will be to find out if fruit flies also have critical period for learning vocal cues and if so, how it is regulated at the molecular and neural levels. A better understanding of how fruit flies learn and discriminate sounds may bridge knowledge gaps in research using humans and other mammals.

DOI: https://doi.org/10.7554/eLife.34348.002

copulation (*Bennet Clark and Ewing, 1969*; *von Schilcher, 1976a*; *von Schilcher, 1976b*). An artificial pulse song also increases sexual behavior in males, even without the presence of females, stimulating 'chaining behavior', in which males chase each other and form male-male chains (*Crossley et al., 1995*; *Yoon et al., 2013*). This chaining behavior presumably arises from the increase of sexual arousal, which induces a male to join the courtship strived by other nearby males (*Eberl et al., 1997*). Intriguingly, the quality of the pulse song affects sexual arousal. The temporal gap between the pulses in the pulse song, namely the inter-pulse interval (IPI), differs among sibling *Drosophila* species (*Cobb et al., 1989*; *Ewing and Bennet-Clark, 1968*; *Ewing and Manning, 1967*) and is thought to be the crucial parameters for sexual arousal and species recognition. Indeed, *D. melanogaster* males prefer the pulse song with a certain range of IPIs including 35 ms, the mean IPI of this species (*Yoon et al., 2013*; *Zhou et al., 2015*). This bias towards the conspecific pulse song raises a question of how IPI selectivity is formed, making the fruit fly a simple model to investigate the mechanism underlying sound perception.

Fruit flies detect sound with antennal ears and, specifically, with mechanosensory neurons of Johnston's organ (JO) (*Kamikouchi et al., 2009*). Regarding the two key features of *Drosophila* pulse song, intra-pulse frequency (IPF) and IPI, the antennal ear is mechanically tuned to detect the conspecific IPF, and the brain is hypothesized to process the conspecific IPI (*Riabinina et al., 2011*). Recently an auditory pathway to perceive the pulse song that underlies the mating decision was delineated in *Drosophila* males. This pathway includes mechanosensory neurons in JO (JO neurons), aPN1 neurons (also known as AMMC-B1 neurons), vPN1 neurons, and pC1 neurons

(*Kamikouchi et al., 2009*; *Vaughan et al., 2014*; *Zhou et al., 2015*). In males, the pC1 cluster includes the courtship command-like P1 neurons. Multi-stage transformations by neurons in this auditory pathway refine the perception of IPIs until the response of the pC1 neurons matches the behavioral response to songs with different IPIs. These studies illustrate how the tuning towards the conspecific song with 35 ms IPI can be achieved, and raise the question of how this IPI preference emerges. Although it is traditionally believed that the courtship behavior of flies is innate (*Auer and Benton, 2016*; *Baker et al., 2001*; *Hall, 1994*), the programmed courtship machinery is susceptible to variables in development such as sleep deprivation (*Kayser et al., 2014*), social isolation (*Kim et al., 1998*; *Pan and Baker, 2014*) and juvenile social experience (*McRobert and Tompkins, 1988*). Pioneering studies on zebra finches (*Chen et al., 2017*; *Cousillas et al., 2006*; *Woolley et al., 2010*; *Yanagihara and Yazaki-Sugiyama, 2016*) and bats (*Razak et al., 2008*) suggest that auditory selectivity in these animals developed in an experience-dependent manner. Accordingly, we hypothesized that in young flies, IPI preference might also be refined by the experience of songs from nearby males, which might modulate the partner selection in sexual behaviors.

In this study, we examined whether *Drosophila* IPI selectivity was tuned by the auditory experience. Based on the sexual behaviors of males and females upon song playback, we established a new behavioral paradigm in which the flies were exposed to specific sound patterns for long periods before their IPI preference was evaluated. Surprisingly, we found that the experience of conspecific song, but not heterospecific song, tuned IPI perception in both males and female flies. Furthermore, we found that this experience-dependent IPI tuning relied on GABA synthesis, and that the ionotropic GABA$_A$ receptor of pC1 neurons gated IPI tuning in females. Our discovery establishes a new and simple system to study how the experience-dependent auditory plasticity is incorporated into higher-order integration center to modulate sexual behaviors at the molecular and cellular levels.

## Results

### Experience-dependent tuning of IPI preference in male fruit flies

In *Drosophila melanogaster*, IPIs ranging from 35 ms to 75 ms induce the sexual behavior of males vigorously (*Yoon et al., 2013*). Since the mean IPI of the courtship song in *D. melanogaster* is about 35 ms (*Cowling and Burnet, 1981*), it seems noteworthy that 75 ms IPI, which is out of the *melanogaster* IPI range (*Arthur et al., 2013*) and likely comes from another *Drosophila* species (for example, an evolutionarily far species *Drosophila rosinae* in *fasciola* subgroup) (*Costa and Sene, 2002*), induces sexual behavior as strongly as 35 ms IPI. We noticed that male flies that showed similar levels of response to both 35 ms and 75 ms IPI songs had been wing-clipped soon after eclosion and thus lacked experiences of wing-emitted sound (*Yoon et al., 2013*). Because *Drosophilid*s gather in groups in feeding sites (*Powell, 1997*), we reasoned that flies probably had experiences of the courtship songs of other males in social interactions, and tested how the auditory experience affected the IPI selectivity.

To evaluate how the experience of wing-emitted sound from other males affects later acoustic preference, we measured the chaining behavior of males that were reared for five to six days in the following three conditions: (1) grouped flies without wings, (2) grouped flies with intact wings, and (3) single-reared flies with intact wings. The wings of males in the latter two groups were clipped only one day before the chaining test. For the chaining test, we used two types of artificial pulse songs: 35 ms IPI and 75 ms IPI songs to represent conspecific and heterospecific songs, respectively. Consistent with our previous report (*Yoon et al., 2013*), flies grouped without wings responded strongly to both conspecific and heterospecific songs (*Figure 1A*). In contrast, flies grouped with wings preferred conspecific over heterospecific song (*Figure 1B*). This selective response was not observed in single-reared flies with wings (*Figure 1C*). Together, these results indicate that the presence of other males with wings is required to shape the IPI preference in males.

To investigate whether the prior sound experience modifies the IPI selectivity, we established a training procedure containing a training session and a subsequent test session (*Figure 2A and B*). In the training session, we exposed wing-clipped single males to conspecific or heterospecific artificial song for 6 days after eclosion, which served as 'auditory experience' to flies. Naïve flies were also prepared in the same manner as experienced flies except for the exposure to the training sound. In the test session, we monitored their behavioral performance using chaining test. Conspecific song

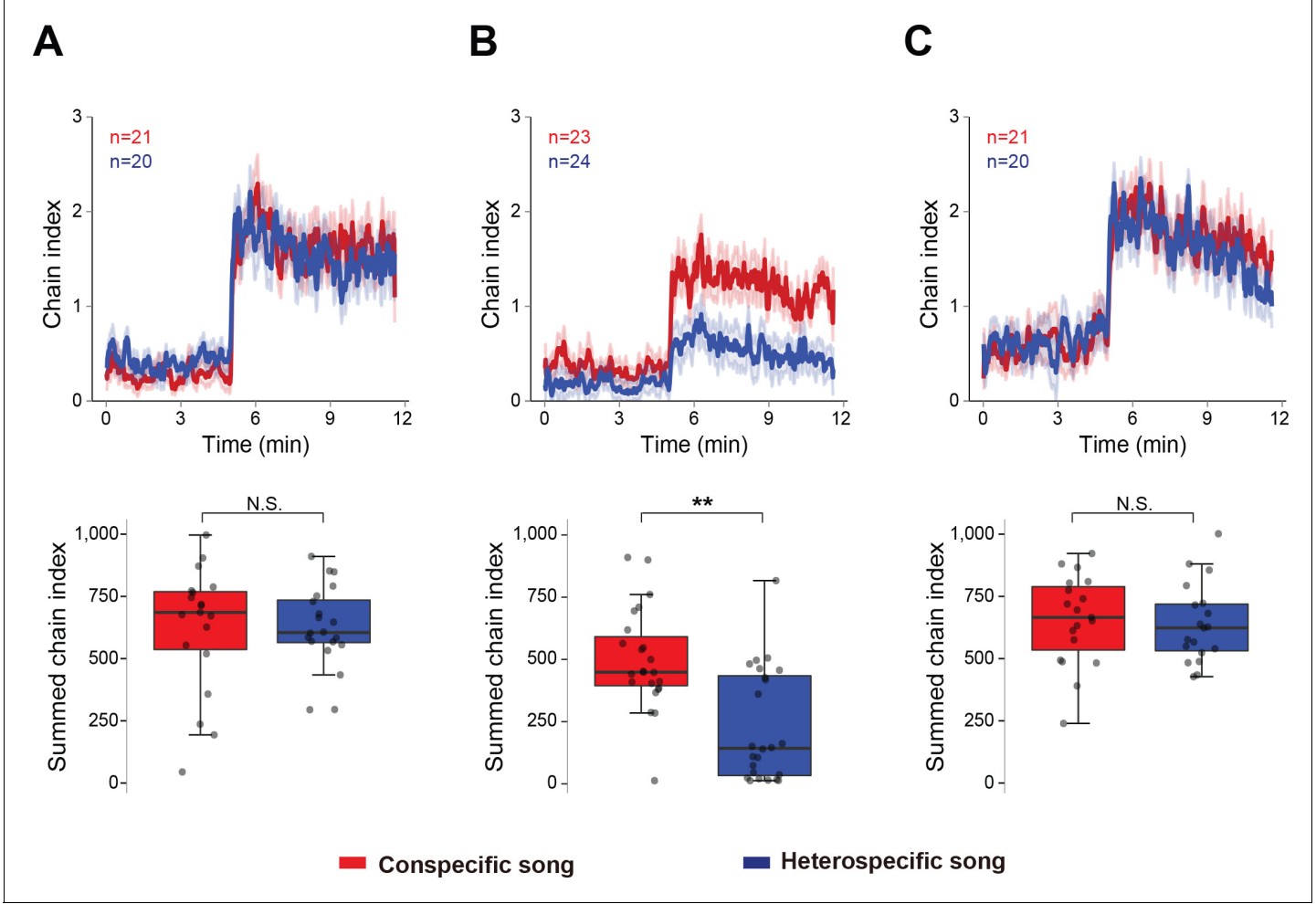

**Figure 1.** Social interaction shapes the preference to the song. Chaining response of naïve male flies that were housed in different experimental conditions, grouped without wings (**A**), grouped with intact wings (**B**), and single-reared with intact wings (**C**). The time-courses of the chain index in response to playback of conspecific song (red) and heterospecific song (blue) are shown. Sound playback starts at 5 min. The bold line and ribbon represent the average value and standard error, respectively. The box plot shows the summed chain index between 5 min and 11.5 min. Boxplots display the median of each group with the 25th and 75th percentiles and whiskers denote 1.5x the inter-quartile range. N.S., not significant, p>0.05; **p<0.01; Mann-Whitney U test. n, number of behavioral chambers examined.

DOI: https://doi.org/10.7554/eLife.34348.003

induced a strong chaining behavior of males in both naïve and experienced groups, irrespective of the training sound (*Figure 2C*). In contrast, heterospecific song induced a strong chaining behavior in naïve but not in experienced flies when flies were trained with conspecific song (*Figure 2D*, red line). Flies trained with heterospecific song retained their response to the heterospecific song (*Figure 2D*, orange line). These results indicate that male flies selectively diminish the response to heterospecific song only after having experienced conspecific song.

## Experience-dependent tuning of IPI preference in female fruit flies

Females decide whether to mate with courting males (*Dickson, 2008*). To test whether the mating decision of females could also be tuned by a prior auditory experience, we probed song effects on copulation behavior (*Figure 3A*). First, we paired naïve females with naïve wing-clipped males to confirm the IPI selectivity in promoting copulation as reported (*Bennet Clark and Ewing, 1969*). Compared with the test condition without sound playback, either conspecific (35 ms) or heterospecific (75 ms) song playback promoted copulation significantly (*Figure 3B*). Both songs promoted copulation equally, showing that naïve females had no selectivity between these two songs. In

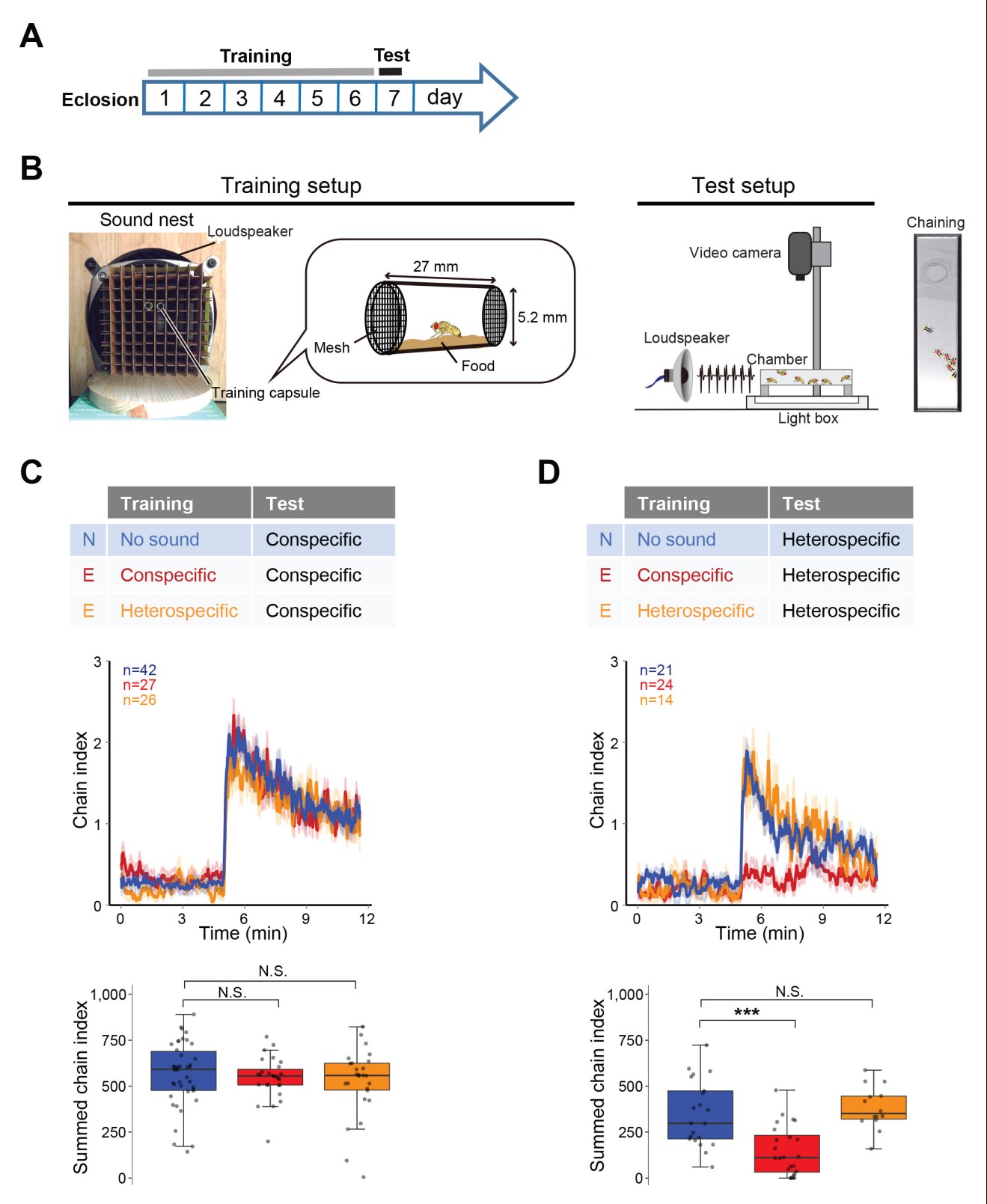

**Figure 2.** Fine-tuned song response of males after the training. (**A**) Protocol for the training and test sessions. (**B**) Setup for the training and chaining test. In the training session, single-housed male flies were exposed to a training song for the first 6 days after eclosion. In the test session, song was delivered from a loudspeaker. Appropriate song typically drove the male flies to form male-male chains (chaining). Males in a chain are marked with red dots. (**C, D**) Chaining response to the conspecific song (**C**) or heterospecific song (**D**) after training. N, naïve group with no sound training (blue); E,

*Figure 2 continued on next page*

*Figure 2 continued*
experienced group with conspecific song training (red) or heterospecific song training (orange). The way to show the time courses of chaining behavior and the boxplot is similar to that depicted in *Figure 1*. N.S., not significant, p>0.05; ***p<0.001; Mann-Whitney U test versus naïve group.
DOI: https://doi.org/10.7554/eLife.34348.004

contrast, playback of songs with shorter (15 ms) or longer (105 ms) IPIs did not promote copulation. These results are consistent with previous findings that only songs with certain IPIs in a specific range promoted copulation(*Bennet Clark and Ewing, 1969*).

Then we tested whether previous sound experience affects female copulation behavior. We trained the females with conspecific or heterospecific song, in the same way as for males (*Figure 2B*), and then tested the female receptivity to a mute male with song playback (*Figure 4*). To examine the song training effect on females, naïve or trained females were paired with naïve wild-type males that were wing-clipped for copulation test. With playback of conspecific song, females accepted mating with mute males regardless of the song experience during the training session (*Figure 4A*). In contrast, with heterospecific song playback, the copulation rate dramatically decreased in females trained with the conspecific song (*Figure 4B*, red line). Training with

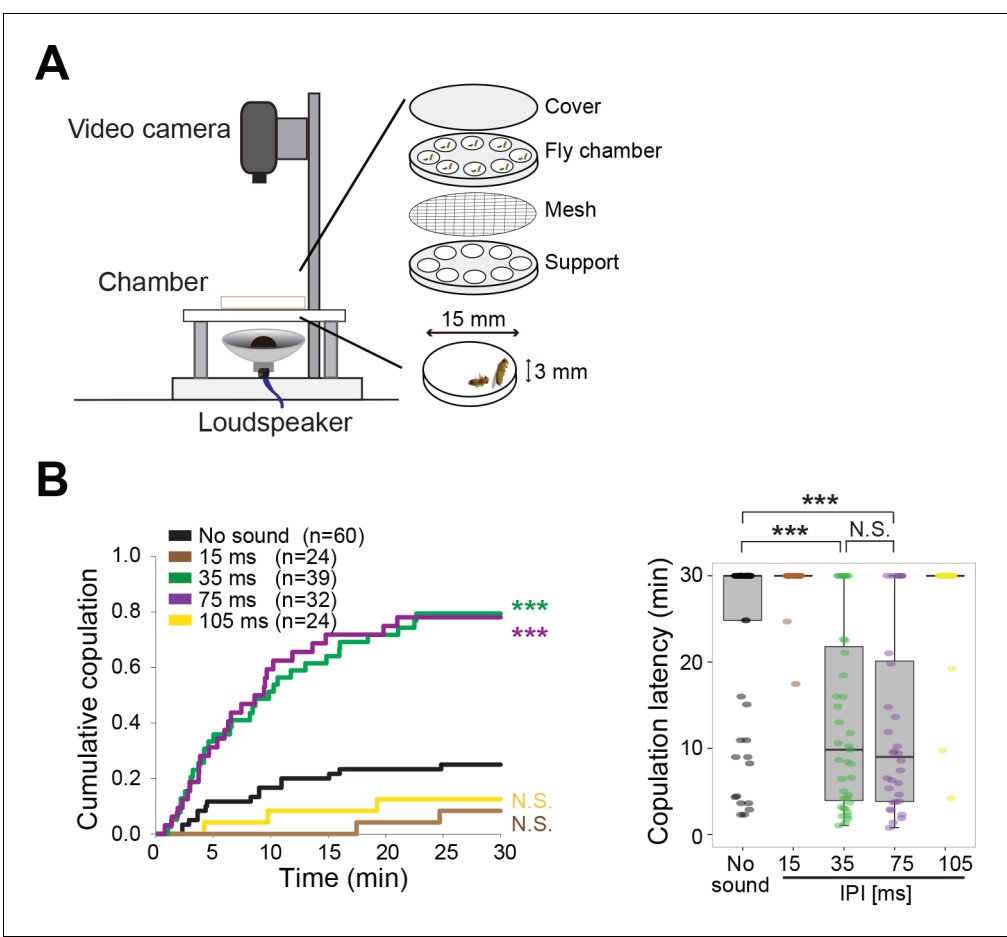

**Figure 3.** Playback of pulse song promotes copulation in wild-type fly pairs. (A) Setup for song-induced copulation test. (B) Cumulative copulation rate and copulation latency with playback of artificial pulse songs of different inter-pulse interval (IPI). Copulation latency represents the latency to accept copulation in the 30 min observation period. Boxplots display the median of each group with the 25th and 75th percentiles and whiskers denote 1.5x the inter-quartile range. N.S., not significant, p>0.05; ***p<0.001; Log rank test versus no sound group (left panel); Kruskal–Wallis test followed by Scheffe's test (right panel). n, number of fly pairs examined.
DOI: https://doi.org/10.7554/eLife.34348.005

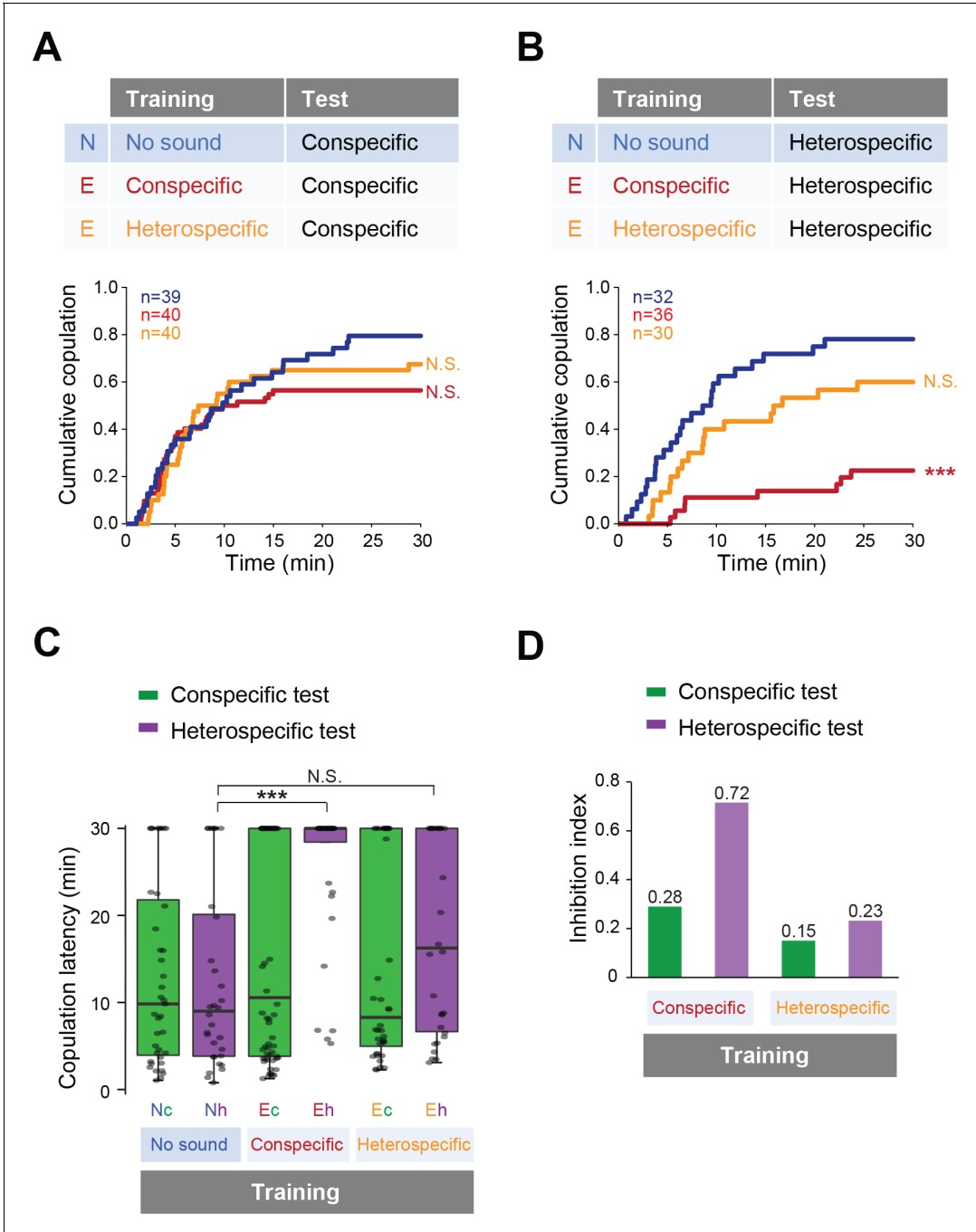

**Figure 4.** Fine-tuned song response of females after the training. (A, B) Cumulative copulation rate in the conspecific song test (A) or heterospecific song test (B) after training females. Naïve group (no sound during training) and experienced groups (trained with conspecific song or heterospecific song) are shown. The color code is the same with that in *Figure 2*. N, naïve; E, experienced. (C) Copulation latencies of females under playback of conspecific song (green bars) or heterospecific song (purple bars). Nc and Nh, naïve flies tested with conspecific and heterospecific songs, respectively. Ec and Eh, experienced flies tested with conspecific and heterospecific songs, respectively. (D) Inhibition index under playback of conspecific song (green bars) or heterospecific song (purple bars) after training of conspecific song or heterospecific song. Inhibition index = (copulation ratio[Naïve] - copulation ratio[Experienced])/copulation ratio[Naïve]. N.S., not significant, p>0.05; ***p<0.001; Log rank test versus naïve group (A, B); Kruskal–Wallis test versus naïve group (C).

DOI: https://doi.org/10.7554/eLife.34348.006

heterospecific song did not affect the receptivity to the heterospecific song (*Figure 4B*, orange line). Both the significant increase of the copulation latency (*Figure 4C*) and the highest inhibition index of the copulation rate (*Figure 4D*) supported the conclusion that training of the conspecific song reduced female acceptance during the heterospecific song test.

Taken together, previous experience of the conspecific song renders females more selective about the song when deciding to accept mating. Apparently, prior experience of the conspecific song fine-tunes the selectivity of the sound-evoked behavioral responses of both males and females, while prior experience of the heterospecific song does not.

## Experience-dependent IPI tuning requires GABA synthesis

We next sought to identify the mechanism of this experience-dependent tuning of auditory behavior. In mammals, auditory experience governs the maturation of GABAergic inhibition that tunes the perception of sound in the auditory cortex (*Dorrn et al., 2010*). Thus we asked whether GABA signaling was involved in the auditory plasticity that we found, by testing the receptivity of female flies with reduced GABA synthesis. We knocked down *Glutamic acid decarboxylase 1* (*Gad1*), a gene encoding the major GABA synthesis enzyme, in putative GABAergic neurons (*Gad1-GAL4 > UAS-Gad1 RNAi*; see Materials and methods for fly strains) in females, and trained them with conspecific or heterospecific song. The copulation tests with conspecific song playback revealed that both *Gad1* knockdown and control (*Gad1-GAL4 > RNAi* background $w^{1118}$) females in experienced groups responded to conspecific song as strongly as naïve females, irrespective of training experience (*Figure 5A*).

In contrast, when we used heterospecific song in the tests, *Gad1* knockdown females showed two phenotypes different from the control group (*Figure 5B*). The first phenotype came after training of conspecific song (*Figure 5B*, red lines); while control females reduced receptivity like wild-type females (*Figure 5B*, right), receptivity of *Gad1* knockdown females stayed at the same level as in naïve females (*Figure 5B*, left). This result suggests the necessity of GABA in this experience-dependent IPI tuning. The second phenotype appeared after heterospecific song training (*Figure 5B*, orange lines); *Gad1* knockdown flies decreased their copulation rate dramatically when compared with naïve flies (*Figure 5B*, left), whereas control flies (*Figure 5B*, right) and wild-type flies (*Figure 4B*) did not. These results demonstrate that although the response to the conspecific song in females was neither interrupted by *Gad1* knockdown nor by training (*Figure 5A*), the response to heterospecific song was vulnerable to *Gad1* knockdown and training (*Figure 5B*). Training with both conspecific song and heterospecific song might have modified properties of the neural circuit for the processing of heterospecific song. GABA synthesis is necessary to show the plasticity induced by conspecific song training, and to defend against the modulation induced by heterospecific song training as well.

Together, these results prove that GABA synthesis is necessary for the IPI tuning induced by conspecific song training, which is reminiscent of the involvement of GABA in auditory plasticity exhibited in mammals and songbirds (*Dorrn et al., 2010*; *Kotak et al., 2008*; *Yanagihara and Yazaki-Sugiyama, 2016*).

## GABA mediates the experience-dependent plasticity via Rdl receptors in pC1 neurons

P1 neurons, a male-specific subset of pC1 neurons, are the mating command-like neurons that receive multimodal input from olfactory, gustatory, and auditory systems (*Auer and Benton, 2016*). Multimodal sensory information is transmitted to P1 neurons through excitatory and inhibitory pathways to achieve a stringent control of courtship decision-making in males (*Clowney et al., 2015*; *Koganezawa et al., 2016*). In these pathways, GABA transmits inhibitory signals to P1 neurons via GABA$_A$-type Rdl receptors (*Kallman et al., 2015*; *Koganezawa et al., 2016*). Similarly, female pC1 neurons, the counterpart of male pC1 neurons (*Koganezawa et al., 2016*), regulate female receptivity by evaluating sexual signals from males including the courtship song and the male-specific phero-mone cVA (*Zhou et al., 2014*). Under the hypothesis that GABA signaling via Rdl receptors might also regulate female pC1 neurons, we asked whether pC1 neurons in females were the target neurons of GABA that mediates the experience-dependent IPI tuning. We knocked down the expression of *Rdl* by driving *Rdl* RNAi specifically in female pC1 neurons, defined by the intersection of an

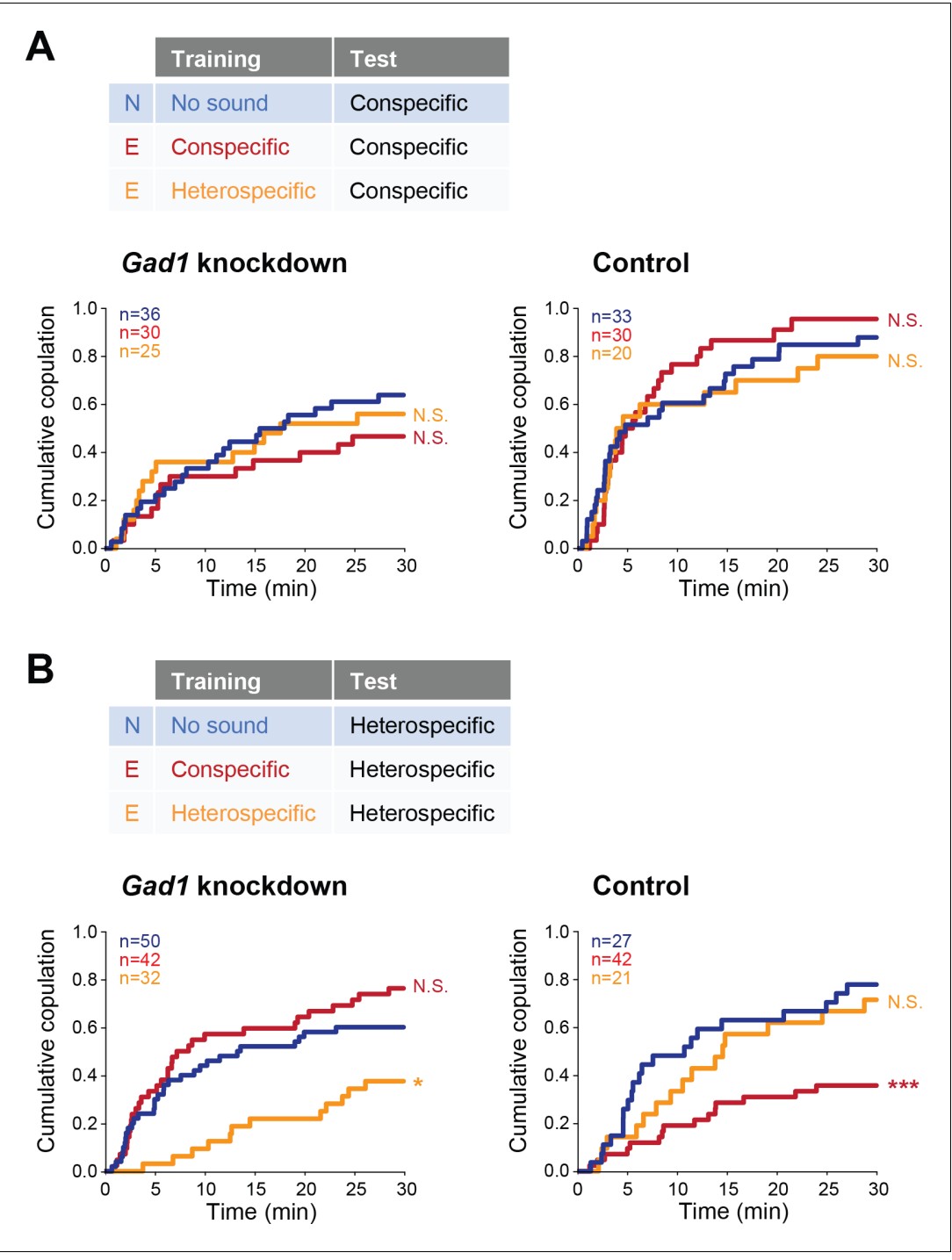

**Figure 5.** Involvement of *Gad1* in the experience-dependent song preference in females. (**A, B**) Cumulative copulation rate in the conspecific song test (**A**) or heterospecific song test (**B**) after training *Gad1* knockdown (left) and control (right) females. Naïve group (no sound training) and experienced groups (conspecific song training and heterospecific song training) are shown. The color code is the same with that in *Figure 2*. N, naïve; E, experienced. N.S., not significant, p>0.05; *p<0.05; ***p<0.001; Log rank test versus naïve group.

DOI: https://doi.org/10.7554/eLife.34348.007

The following figure supplement is available for figure 5:

**Figure supplement 1.** Male *Gad1* knockdown flies responded normally to conspecific courtship song.
DOI: https://doi.org/10.7554/eLife.34348.008

enhancer trap line *NP2631* and *dsx^FLP^* (**Koganezawa et al., 2016**). Consistent with the aforementioned results, in the conspecific song test both *Rdl* knockdown and control females in experienced

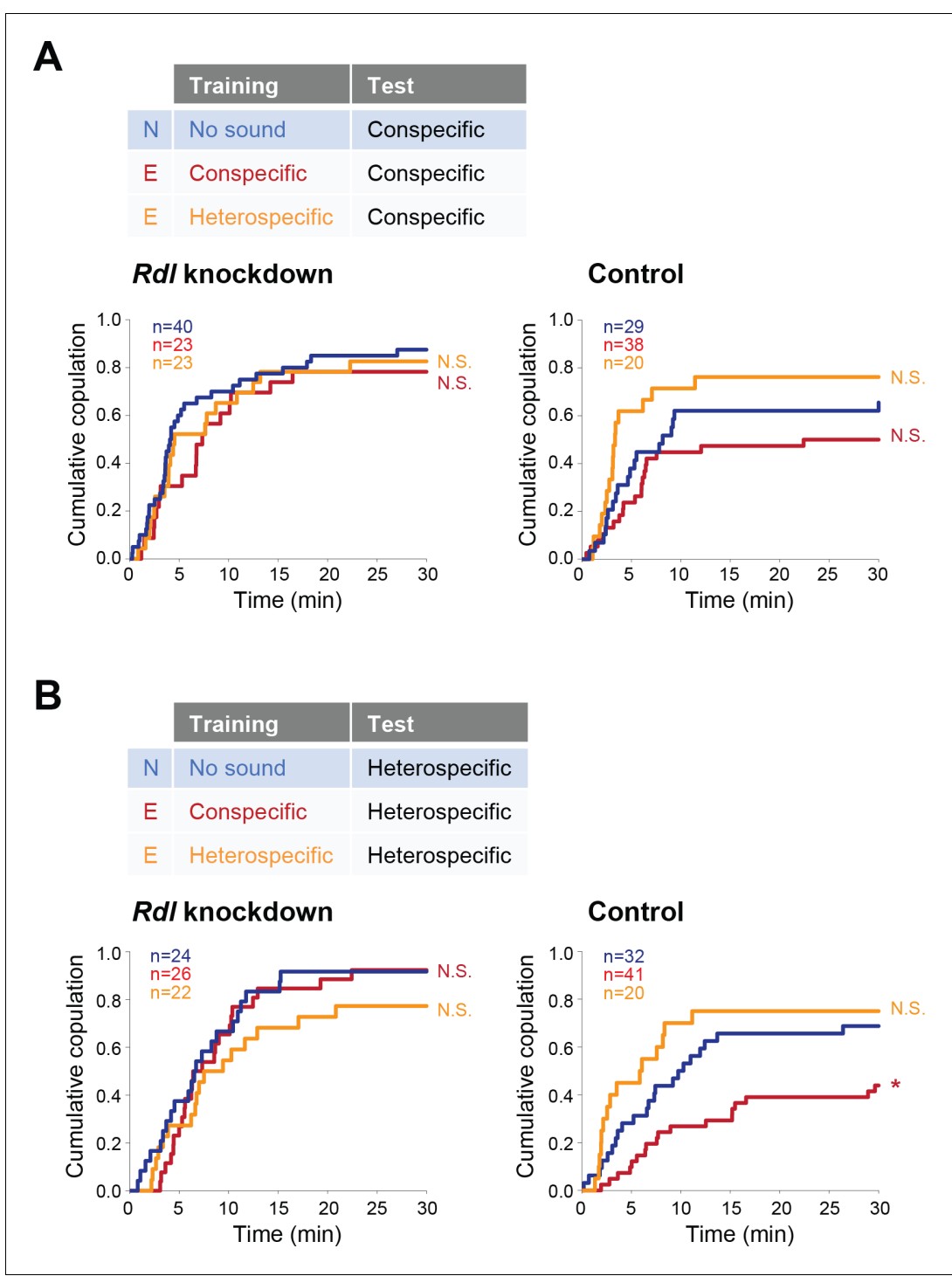

**Figure 6.** Rdl receptors in pC1 neurons modulate the experience-dependent song preference in females. (A, B) Cumulative copulation rate in the conspecific song test (A) or heterospecific song test (B) after training *Rdl* knockdown (left) and control (right) females. Naïve group (no sound training) and experienced groups (conspecific song training or heterospecific song training) are shown. The color code is the same with that in **Figure 2**. N, naïve; E, experienced. N.S., not significant, p>0.05; *p<0.05; Log rank test versus naïve group.
DOI: https://doi.org/10.7554/eLife.34348.009

groups responded similarly as naïve females did, irrespective of training experiences (*Figure 6A*). In the heterospecific song test, however, *Rdl* knockdown females, but not control and wild-type ones, kept the receptivity to the heterospecific song even after training with the conspecific song (*Figure 6B*). Accordingly, knockdown of *Rdl* in pC1 neurons abolishes the experience-dependent tuning of the IPI, indicating that GABA mediates this IPI tuning via GABA$_A$ receptors in pC1 neurons of females.

Interestingly, training with heterospecific song induced no changes in both *Rdl* knockdown and control groups (*Figure 6A and B*). This result contrasts with that in *Gad1* knockdown flies, in which the experience of heterospecific song reduced the female receptivity upon exposure to heterospecific song (*Figure 5B*). Rdl receptors in female pC1 neurons are thus unlikely to be the direct target of GABA signaling to defend against the modulation induced by the training of heterospecific song.

## Discussion

The courtship behavior of *Drosophila melanogaster* provides a simple model to understand how the innate perception of sensory signals is configured to direct the higher cognitive functions. Especially, the perception of auditory signals has attracted much attention from researchers because acoustic communication plays important roles in species identification and reproductive isolation of not only fruit flies, but also other animals such as birds (*Catchpole, 1987*), fishes (*Amorim et al., 2015*), frogs (*Backwell and Jennions, 1993*), and crickets (*Hedwig, 2006*). Here we identify a novel phenomenon revealing the experience-dependent auditory plasticity that shapes sexual preference in fruit flies (*D. melanogaster*). Analogous to the regulatory role of GABA in shaping auditory circuits of zebra finch (*Yanagihara and Yazaki-Sugiyama, 2016*), we demonstrate that GABA signaling also shapes auditory selectivity in flies. We further identify the receptors responsible for this signaling on a small subset of central neurons that mediate the tuning of IPI perception. Our findings document how the experience-dependent mechanism is incorporated with an innate auditory system and accordingly establish the fruit fly, with its abundant molecular-genetic tools, as a powerful model to investigate the mechanisms of auditory plasticity on the molecular and cellular levels.

### Song experience shapes the IPI preference

Temporal pattern of sound is a crucial feature in the communication signals of many animals, such as in bird songs, frog calls, cricket chirps, and human speech (*Pollack, 2001*). Particularly in lower-vertebrates and insects, understanding the simple patterns of sounds used in communication, such as the specific pulse rate, is important in deciphering the meanings of these signals (*Alexander, 1962*; *Bass and McKibben, 2003*; *Schöneich et al., 2015*). Fruit flies use the pulse songs with a species-specific IPI during courtship (*Ewing and Bennet-Clark, 1968*). In this study, we found that the flies' initial wide-ranging IPI preference was refined by early auditory experience. Since the IPI distribution in the recorded natural courtship song is particularly enriched at around 35 ms (*Arthur et al., 2013*), young adult flies are highly likely to be exposed to this conspecific IPI emitted by other males. This experience might tune the IPI preference and predispose partner selection in sexual behavior later in life. Indeed, our results prove that social interaction during early adulthood tunes the IPI preference towards the conspecific IPI (35 ms) (*Figure 1*). This beautiful coordination between innate preference and experience-dependent refinement allows enough flexibility in mating, and reduces the risk of crossbreeding between species, which contributes to species isolation.

We find that only the experience of conspecific song tunes the auditory preference, while the experience of heterospecific song does not. This asymmetric learning of conspecific and heterospecific songs suggests that naïve flies can already distinguish conspecific song from heterospecific song, since only the former is capable of modifying their later preference behavior.

We previously reported that male *D. melanogaster* showed equal behavioral preference towards IPIs between 35 ms and 75 ms (*Yoon et al., 2013*), which were used as conspecific and heterospecific songs in the present study. However, another report showed that male *D. melanogaster* preferentially responded to 35 ms over all other IPIs (*Zhou et al., 2015*). This discrepancy can now be explained by the experimental difference between these two studies, whether the male flies kept in a group have the experience of carrying wings (*Zhou et al., 2015*) or not (*Yoon et al., 2013*). As for how long the necessary experience is, and whether a critical period exists, further study is needed to answer these questions.

## Experience-dependent learning refines the mating preference

Whether nature or nurture plays dominant roles in the formation of animal behavior has been debated for a long time, yet the courtship behavior of *D. melanogaster*, including its underlying sensory perception, has long been recognized to be innate. Numerous empirical evidences have supported the capability of single-reared flies to perform all the courtship steps spontaneously and completely (*Auer and Benton, 2016*; *Baker et al., 2001*; *Hall, 1994*). However, our results reveal that the specific sound experience is necessary to refine the auditory preference in sexual behavior, which for the first time suggests a mechanism of learning in the song discrimination of flies.

In fact, animals in many species learn their mating preferences. One notable example is sexual imprinting, the process whereby mating preferences are affected by learning the species-specific characteristics at a very young age (*Irwin and Price, 1999*). As observed in birds (*Ten Cate, 1999*), fishes (*Kozak et al., 2011*), and sheep and goats (*Owens et al., 1999*), an early period of social interaction with parents or siblings helps the learner discriminate sex and species by learned phenotypic traits, and affects mating preference in the future (*Verzijden et al., 2012*). Here we provide evidence that fruit flies refine the IPI preference by sexual imprinting, which would reinforce reproductive isolation together with innate auditory perception. This sexual imprinting of courtship song is apparently different from the lessons learned from the successful courtship experience (*Saleem et al., 2014*) or unsuccessful courtship attempts (*Griffith and Ejima, 2009*), by which male flies become more competitive over other males, or learn to avoid either mated or heterospecific females. Previous behavioral studies also indicated that social experience in juvenile stage affected adult courtship behaviors of insects. In crickets, juvenile experience of acoustic sexual signals influenced the development of three traits in adult: reproductive tactics, reproductive investment, and body condition (*Bailey et al., 2010*). In fruit flies, young males courted by mature males with intact wings mated significantly faster than those that had been stored alone, suggesting auditory experience in immature stage might affect later courtship (*McRobert and Tompkins, 1988*). Consistent with these observations and going deeper, our study directly demonstrated, with the underlying mechanisms, that auditory experience during the immature stage shaped perception of courtship song, and directed the sexual behavior at the adult stage.

## A new model to study auditory plasticity

Our findings greatly expand the understanding of the experience-dependent auditory plasticity in insects, whose mechanism is consistent with that of mammals and finches. In vertebrates, maturation of excitation-inhibition balance that governs sound perception requires acoustic experience. In rats, developmental sensory experience balances the excitation and inhibition in the primary auditory cortex (A1) (*Dorrn et al., 2010*), whose stereotyped sequential occurrence sharpens spike timing (*Wehr and Zador, 2003*). Hearing loss hinders the maturation of GABAergic transmission mediated by GABA$_A$ receptors in the auditory cortex of gerbils (*Kotak et al., 2008*). In zebra finch, experience-dependent recruitment of GABAergic inhibition in the auditory cortex is necessary to form the memory template of the tutor song (*Yanagihara and Yazaki-Sugiyama, 2016*). In flies, our results also suggest that song experience recruits GABAergic inhibition on the auditory pathway, and the coordination of excitation and inhibition controls

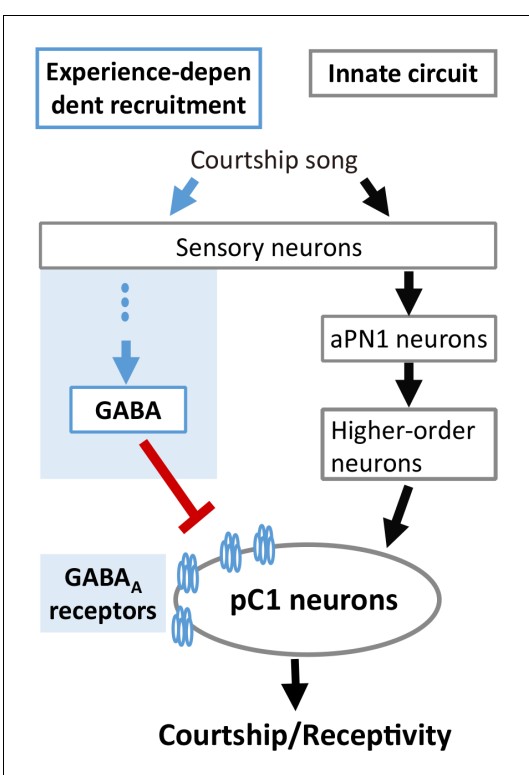

**Figure 7.** A model for experience-dependent tuning of IPI perception in *Drosophila*.
DOI: https://doi.org/10.7554/eLife.34348.010

auditory responses and behavioral output (*Figure 7*). Interestingly, the phenotypes of *Gad1* knockdown in GABAergic neurons and *Rdl* knockdown in pC1 neurons were different when females were tested with heterospecific song (*Figures 5* and *6*). This finding suggests that there are at least two distinct GABAergic pathways to control the experience-dependent auditory plasticity. How these GABAergic pathways are organized cooperatively to shape the IPI preference awaits further analysis.

Interestingly, the combination of excitation and inhibition that modulates the mating decision in flies is not restricted to the auditory system, but is also conserved in olfactory and gustatory systems (*Auer and Benton, 2016*; *Clowney et al., 2015*; *Kallman et al., 2015*). The difference is that the sexual circuitry in the chemosensory modalities is thought to be hard-wired (*Auer and Benton, 2016*; *Hall, 1994*; *Pan and Baker, 2014*), while the inhibition we find in the auditory system matures with experience. Intriguingly, all these inhibitions found in olfactory, gustatory, and auditory pathways function directly on the pC1 neurons, strengthening the role of pC1 neurons as a crucial neural circuit node for multimodal integration (*Auer and Benton, 2016*; *Clowney et al., 2015*; *Kallman et al., 2015*).

The discovery that only the training of conspecific song refines the IPI preference of wild-type flies is reminiscent of vocal learning in zebra finches, which preferentially learn the courtship song of their own species (*Brenowitz and Woolley, 2004*; *Doupe and Kuhl, 1999*). The courtship song preferences in female zebra finches are shaped by the developmental auditory experience (*Chen et al., 2017*), which shares great similarity with that in fruit flies. The IPI in the courtship song of flies resembles the temporal gap between syllables in the finch song, which serves as a 'barcode' for song identity (*Araki et al., 2016*). The auditory study of these two model organisms therefore might complement and enlighten each other in exploring the mechanism of experience-dependent plasticity in conspecific sound perception, potentially contributing to the understanding of language acquisition in humans. Unlike zebra finch, fruit flies rarely see their parents. Our results demonstrate that learning from adolescent peers is sufficient to modulate the perception of IPIs (*Figure 1*). In the natural environment, young flies possibly learn from young flies as well as mature flies. Taken together, our findings open a new research field to use the fruit fly, with its abundant molecular-genetic tools and simple neural circuits, to study the experience-dependent auditory information processing and sensorimotor output, which are challenging to examine at the molecular and cellular levels in zebra finches and primates including humans.

## Materials and methods

### Key resources table

| Reagent type (species) or resource | Designation | Source or reference | Identifiers | Additional information |
|---|---|---|---|---|
| gene (*Drosophila melanogaster*) | *Rdl* | NA | FLYB: FBgn0004244 | |
| gene (*Drosophila melanogaster*) | *GAD1* | NA | FLYB: FBgn0004516 | |
| strain, strain background (*Drosophila melanogaster*) | *Canton-S* | other | | gift from K. Ito |
| strain, strain background (*Drosophila melanogaster*) | *Gad1-GAL4* | PMID: 12408848 | | gift from K. Ito |
| strain, strain background (*Drosophila melanogaster*) | *UAS-Gad1 RNAi* | Vienna Drosophila Resource Center | VDRC ID: 32344; RRID: Fly-Base_FBst0459538 | |
| strain, strain background (*Drosophila melanogaster*) | *w1118* | Vienna Drosophila Resource Center | VDRC ID: 60000 | |
| strain, strain background (*Drosophila melanogaster*) | *UAS-Rdl RNAi* | Bloomington Drosophila Stock Center | BDRC: 52903; RRID: BDSC_52903 | |
| strain, strain background (*Drosophila melanogaster*) | *TRiP RNAi* | Bloomington Drosophila Stock Center | BDRC: 36304; RRID: BDSC_36304 | |
| strain, strain background (*Drosophila melanogaster*) | *tubP>GAL80>; NP2631-GAL4/CyO; dsxFLP/TM2* | PMID: 27185554 | | gift from D. Yamamoto |
| software, algorithm | ChaIN (ver. 3) | PMID: 28701929 | | |

## Experimental animals

*D. melanogaster* was raised on standard yeast-based media at 25°C and in 40% to 60% relative humidity on a 12 hr light/dark cycle. *Canton-S* (Hotta-lab strain, a gift from K. Ito) was used as a wild-type strain. For knockdown experiments, the following transgenic flies were used: *w; Gad1-GAL4* (*Ng et al., 2002*) (a gift from K. Ito), *UAS-Gad1 RNAi* (GD line; RRID: FlyBase_FBst0459538) and its control line *w^1118^* (VDRC ID: 60000) (Vienna *Drosophila* Resource Center), *UAS-Rdl RNAi* (VALIUM20; RRID: BDSC_52903) and its control line *TRiP RNAi* (RRID: BDSC_36304) (Bloomington *Drosophila* Stock Center), and *tubP>GAL80>; NP2631-GAL4/CyO; dsx^FLP^/TM2* (*Koganezawa et al., 2016*) (a gift from D. Yamamoto). Genotypes of flies used for each experiment are listed in *Supplementary file 1*. Flies that were 6 to 7 day after eclosion were used for behavioral tests. The wings of males were clipped on the day of eclosion, unless otherwise noted.

The neurons labeled by *Gad1-GAL4* show essentially consistent distributions with those identified by in situ hybridization against *Gad1* mRNA (*Okada et al., 2009*). Silencing these *Gad1-GAL4* positive neurons in the adult stage did not affect fly survival (*Muthukumar et al., 2014*). The *Gad1* RNAi used in this study was reported to knock down the *Gad1* mRNA level to approximately 60% of wild type (*Jeong et al., 2016*). In our study, no obvious behavioral defects were observed in *Gad1* knockdown flies, and male *Gad1* knockdown flies still responded normally to conspecific courtship song when tested at 7 days after eclosion (*Figure 5—figure supplement 1*). The efficacy of *UAS-Rdl RNAi* has been demonstrated (*Franco et al., 2017*; *Koganezawa et al., 2016*).

## Male experiment without the training session

Virgin males were collected within 10 hr after eclosion, and then housed in three different conditions: (1) grouped without wings, (2) grouped with intact wings, and (3) single-reared with intact wings. Flies housed in the first condition (grouped without wings) were prepared as described previously (*Yoon et al., 2013*). In brief, their wings were clipped with forceps during brief anesthesia on ice soon after eclosion and the males were kept in a male-only group of 6 to 8. Flies housed in the second (grouped with intact wings) and third (single with intact wings) conditions were kept with intact wings for 5 to 6 days, either in a group of 6 to 8 male flies or singly. Only one day before the test, the wings of flies housed in the second and third conditions were also clipped. The chaining behavior of all the males housed in three conditions was tested 6 to 7 days after eclosion.

## Training

A protocol for the training session is described in more detail at Bio-protocol (*Li et al., 2018*). Training session started on the day of eclosion. Adult virgin males and females were collected within 8 hr after eclosion under anesthesia on ice, and the wings of males were clipped. Each fly, whether a male or a female, was introduced gently to a training capsule and placed in front of a loudspeaker (FF225WK, FOSTEX, Foster Electric Company, Tokyo, Japan). As experienced group, flies were continuously exposed to one particular training song for 6 days of training (*Figure 2A*). Training song was an artificial pulse song comprised of the repetition of 1 s pulse burst and a subsequent 2 s pause, in which the pulses in the pulse burst had an IPI of 35 ms ('conspecific song') or an IPI of 75 ms ('heterospecific song') (*Yoon et al., 2013*). Intrapulse frequency (IPF) of both IPI songs was set to be 167 Hz. As naïve group, flies were placed in front of the loudspeaker for 6 days after eclosion but not given any sound exposure.

During the training session, each fly was accommodated singly in a training capsule. A training capsule was made of a glass tube cut out from a Pasteur pipette, two pipette tips, mesh and mending tape (*Figure 2B*). Pipette tips, whose volumes are 1 ml, were cut to make the larger ends about 20 mm long. Two of these 20 mm pieces were hooked to a glass tube at its both ends. The size of a glass tube was about 27 mm long, with the internal diameter of 5.2 mm and the external diameter of 6.5 mm. Both exits of the glass tube were sealed with a piece of mesh stocking (made of nylon and polyurethane), which allowed free passage of air but not the fly. A thin layer of fly food, standard *Drosophila* yeast-based medium, was paved at the bottom of the glass tube. The food in each capsule was renewed every 36 hr.

Training capsules were placed within latticework of a container, named a 'sound nest' (*Figure 2B*). One of the mesh-ends of each training capsule faced the loudspeaker, so that sound could be delivered to each chamber with minimal disturbance. The distance between loudspeaker

and the near end of the training capsules was 24 mm. All the setups for the training were placed into a soundproof box (W450 mm × L450 mm × H450 mm).

Sound playback was controlled by the Windows Media Player on a tablet PC (Windows 8.1, Diginnos DG-D08IWB, Dospara, Tokyo, Japan), and delivered by a loudspeaker with a digital power amplifier (Lepai LP-2020A + NFJ Edition, Bukang Electrics, Jieyang, China). The mean baseline-to peak amplitude of sound particle velocity was 8.6 mm/s when measured at the near end of the training capsules, and 6.6 mm/s at the far end of the training capsule. The sound particle velocity was identical for all training sounds.

After the 6 day training, male flies were collected into a group of seven without anesthesia, and transferred to a vial containing fly food. Female flies were still kept in the training capsules singly without sound playback until the copulation test. After one-night rest without any sound playback, all flies (7 days after eclosion) were subjected to the behavioral tests in the next morning (ZT 0–3) (*Figure 2A*).

## Behavioral tests

A protocol for the test session is described in more detail at Bio-protocol (*Li et al., 2018*).

### Male-male chaining test

For males, the sound-evoked chaining test was performed as described (*Yoon et al., 2013*) (*Figure 2B*). Six flies were loaded into one lane of an acoustic behavior chamber (*Inagaki et al., 2010*) and placed in front of a loudspeaker with a distance of about 11 cm. As the test song, the artificial pulse song with 35 ms IPI or 75 ms IPI as used in the training session was delivered from a loudspeaker with an amplifier (Lepai LP-2020A + NFJ Edition, Bukang Electrics, Jieyang, China). Mean baseline-to peak amplitude of its particle velocity was 9.2 mm/s (*Ishikawa et al., 2017*). The flies' contour was outlined by a backlit LED light box (ComicMaster Tracer, Too Marker Products, Tokyo, Japan), and captured by a monochrome camera (Himawari GE60, Library, Tokyo, Japan) with a zoom lens (Lametar 2.8/25 mm, Jenoptik GmbH, Jena, Germany). Flies were not exposed to sound for 5 min, and then exposed to an acoustic stimulus that lasted for 6.5 min. The recorded video was then down-sampled to 1 Hz and analyzed off-line using ChaIN method (*Yoon et al., 2013*). We measured the number of only the follower flies in chains as the chain index using ChaIN version 3 (*Ishikawa et al., 2017*), which is available at http://www.bio.nagoya-u.ac.jp/~NC_home/chain_E.html. The chain index between 5 min and the end of the sound playback were summed for comparison (summed chain index).

### Female copulation test

For females, their receptivity was evaluated by the time course of cumulative copulation and the latency to accept copulation. To monitor the training effect on females, we paired naïve or trained females with the naïve wild-type males (7 day old, wings clipped). The test chamber, made of plexiglass, was made up by eight circular chambers (15 mm diameter, 3 mm depth) with their bottom covered with mesh for sound penetration (*Figure 3A*). A pair of female and male flies was gently aspirated into one of the eight chambers without anesthesia. A pulse song was delivered to flies by a loudspeaker (Daito Voice AR-10N, Tokyo Cone Paper MFG. Co. Ltd. Saitama, Japan) placed 3.9 cm underneath the chambers. The sound particle velocity was 9.2 mm/s. Song playback was started at the same time as video recording was started. Behaviors of flies were recorded for 30 min with a web camera (Logicool HD Webcam C270, Tokyo, Japan). Copulation timing was analyzed manually from the video playback. Inhibition index = (copulation ratio$^{\text{Naïve}}$ - copulation ratio$^{\text{Experienced}}$)/copulation ratio$^{\text{Naïve}}$.

## Statistical analysis

Statistical analysis was performed with R (version 3.0.3). Mann-Whitney U test (two-tailed) was used to compare two groups of samples in the chaining behavior. Kaplan-Meier curves were generated using R and Log rank test was performed to compare females' accumulative copulation rate between two groups in the copulation tests. The Kruskal–Wallis test (two-tailed) followed by Scheffe's test was used to compare the copulation latency. The detailed statistical results are shown in *Supplementary file 2*. The boxplot was drawn with ggplot2 package of R. Boxplots display the

median of each group with the 25th and 75th percentiles and whiskers denote 1.5x the inter-quartile range.

## Acknowledgements

This work was supported by Grant-in-Aid for Scientific Research (B) (Grants 16H04655 to AK), the Grants-in-Aid for Scientific Research on Innovate Areas 'Memory dynamism' (Grant 25115007 to AK), Challenging Research (Exploratory) (Grant 17K19450 to AK), and Grant-in-Aid for Scientific research (C) (Grant 15K07147 to HI) from the Ministry of Education, Culture, Sports, Science, and Technology, Japan, and Inamori Foundation Research Grant (HI). We are grateful to Dr. Kei Ito, Dr. Daisuke Yamamoto, the Bloomington *Drosophila* Stock Center (BDSC), and Vienna *Drosophila* Resource Center (VDRC) for fly strains. We also thank Dr. You Young-Jai, Dr. Charles Yokohama, Dr. Yoichi Oda, Dr. Leon Avery, Dr. Kazuhiro Wada, Dr. Tsunehiko Kohashi, Dr. Yuki Ishikawa, and Dr. Nao Morimoto for helpful comments and discussions, and Miki Kuno, Yumi Maki, and Yuki Ishikawa for fly maintenance.

## Additional information

### Funding

| Funder | Grant reference number | Author |
|---|---|---|
| Ministry of Education, Culture, Sports, Science, and Technology | Grant-in-Aid for Scientific Research (B) (16H04655) | Azusa Kamikouchi |
| Ministry of Education, Culture, Sports, Science, and Technology | Challenging Research (Exploratory) (17K19450) | Azusa Kamikouchi |
| Ministry of Education, Culture, Sports, Science, and Technology | Grants-in-Aid for Scientific Research on Innovate Areas | Azusa Kamikouchi |
| Ministry of Education, Culture, Sports, Science, and Technology | Grant-in-Aid for Scientific research (C) (15K07147) | Hiroshi Ishimoto |
| Inamori Foundation | Research Grant | Hiroshi Ishimoto |

The funders had no role in study design, data collection and interpretation, or the decision to submit the work for publication.

### Author contributions

Xiaodong Li, Conceptualization, Investigation, Methodology, Writing—original draft; Hiroshi Ishimoto, Supervision, Funding acquisition, Methodology, Writing—review and editing; Azusa Kamikouchi, Conceptualization, Resources, Supervision, Funding acquisition, Investigation, Writing—review and editing

### Author ORCIDs

Xiaodong Li  https://orcid.org/0000-0002-2746-2809
Azusa Kamikouchi  http://orcid.org/0000-0003-1552-6892

### Decision letter and Author response

Decision letter https://doi.org/10.7554/eLife.34348.015
Author response https://doi.org/10.7554/eLife.34348.016

## Additional files

**Supplementary files**

• Supplementary file 1: Genotypes. The genotypes used in figures are as follows. In *Figures 5* and *6*, the genotypes of females are listed, while the paired males are always wild type.
DOI: https://doi.org/10.7554/eLife.34348.011

• Supplementary file 2: Statistical results. The detailed statistical results in each figure are listed. N.S., not significant, p>0.05; *p<0.05; **p<0.01; ***p<0.001.
DOI: https://doi.org/10.7554/eLife.34348.012

• Transparent reporting form
DOI: https://doi.org/10.7554/eLife.34348.013

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
