## [Decision Letter]

Thank you for submitting your article "Auditory Experience Controls the Maturation of Song Discrimination and Sexual Response in *Drosophila*" for consideration by *eLife*. Your article has been reviewed by two peer reviewers, and the evaluation has been overseen by Leslie Griffith as Reviewing Editor and Eve Marder as the Senior Editor. The following individuals involved in review of your submission have agreed to reveal their identity: Aki Ejima (Reviewer #2); Daniel F Eberl (Reviewer #3).

The reviewers have discussed the reviews with one another and the Reviewing Editor has drafted this decision to help you prepare a revised submission.

Summary:

This paper provides the first evidence that in *Drosophila*, as in birds and mammals, neural plasticity plays a role in courtship song preferences. It had long been believed that courtship song preferences were completely innate. The authors show that, while the ability to hear and respond to courtship song is innate, both males and females exposed to certain variations of song parameters during early adulthood modulate their preferences and behavior. This modulation is dependent on GABA signaling, with a major integration center in the pC1 neurons of the auditory neural circuit, which also integrate chemosensory information. This work is major advance in the understanding of how auditory information is processed and integrated into the fly's behavior.

Essential revisions:

1) The authors should cite and discuss the previous works, McRobert and Tompkins 1988 and Bailey et al., 2010, which demonstrated that auditory experience at young stage modified courtship performance in flies and crickets.

McRobert SP, Tompkins L. (1988). Two consequences of homosexual courtship performed by *Drosophila melanogaster* and *Drosophila* affinis males. Evol. 42, 1093-1097. (Hirsch and Tompkins 1994 Neurobiology for review)

Bailey NW, Gray B, Zuk M. (2010). Acoustic experience shapes alternative mating tactics and reproductive investment in male field crickets. Curr. Biol. 20(9) 845-849.

2) The practice of excluding data is fraught with peril. The authors state in Materials and methods that they 'indicate outliers by Xs' in figures. Looking at the figures in question (4C and 3B), it seems that in some cases, close to 1/3 of the data points are called outliers! This does not look appropriate. The variance of the conditions with excluded data is much lower than the conditions without exclusion. This seems arbitrary at best. If the statistics were calculated after exclusion, they are really not reflecting the true variance of the phenomenon.

It is hard to imagine what the criterion for exclusion was. In the case of Figure 3B they are excluding control flies that had a normal latency. How could these possibly be outliers?

The authors need to explain this. If the paper is to be published they need to address the issue of outliers in Materials and methods:

A) They need to have a set of criteria for exclusion that are explained and justified in Materials and methods.

B) That set of criteria must be applied to *all* the data, not just the data sets the authors find difficult to deal with.

---

## [Author Response]

Essential revisions:1) The authors should cite and discuss the previous works, McRobert and Tompkins 1988 and Bailey et al., 2010, which demonstrated that auditory experience at young stage modified courtship performance in flies and crickets.McRobert SP, Tompkins L. (1988). Two consequences of homosexual courtship performed by Drosophila melanogaster and Drosophila affinis males. Evol. 42, 1093-1097. (Hirsch and Tompkins 1994 Neurobiology for review)Bailey NW, Gray B, Zuk M. (2010). Acoustic experience shapes alternative mating tactics and reproductive investment in male field crickets. Curr. Biol. 20(9) 845-849.

Thanks for the insightful suggestions. We are pleased to cite these studies and they have been discussed in the Discussion part of the revised manuscript (subsection “Experience-dependent learning refines the mating preference”, last paragraph). We also cite McRobert and Tompkins’ paper in the Introduction (third paragraph).

2) The practice of excluding data is fraught with peril. The authors state in Materials and methods that they 'indicate outliers by Xs' in figures. Looking at the figures in question (4C and 3B), it seems that in some cases, close to 1/3 of the data points are called outliers! This does not look appropriate. The variance of the conditions with excluded data is much lower than the conditions without exclusion. This seems arbitrary at best. If the statistics were calculated after exclusion, they are really not reflecting the true variance of the phenomenon.It is hard to imagine what the criterion for exclusion was. In the case of Figure 3B they are excluding control flies that had a normal latency. How could these possibly be outliers?The authors need to explain this. If the paper is to be published they need to address the issue of outliers in Materials and methods:A) They need to have a set of criteria for exclusion that are explained and justified in Materials and methods.B) That set of criteria must be applied to all the data, not just the data sets the authors find difficult to deal with.

We did not exclude any data for statistical analysis even if these are represented as outliers in figures. The outliers are defined as data points that are located outside the fences of the boxplot (outside 1.5 times the interquartile range above the upper quartile and below the lower quartile).

Consistent with similar researches, the copulation latency of the fly pairs that did not copulate within 30 min was set to be 30 min. In the first column of Figure 3B and the fourth column of Figure 4C, majority of the fly pairs did not copulate within 30 min (44/60 in the first column of Figure 3B; 28/36 in the fourth column of Figure 4C), which were densely plotted at the ceiling of the graphs. This is why data points that represent successful copulation were marked as outliers. We thought to mark these outliers in the boxplot would remind people of the fact that most fly pairs did not copulate, so we marked these outliers in the boxplot while included all the data points in the statistical comparison. The same standard was applied to all the data in this study.

To avoid confusion, in the revised manuscript we did not denote the data points located outside the 1.5x interquartile range as outliers. And still all the data points were considered in statistics.